# One-Step Preparative Separation of Fucoxanthin from Three Edible Brown Algae by Elution-Extrusion Countercurrent Chromatography

**DOI:** 10.3390/md20040257

**Published:** 2022-04-07

**Authors:** Danting Chen, Yating Jin, Di Hu, Jing Ye, Yanbin Lu, Zhiyuan Dai

**Affiliations:** Key Laboratory of Aquatic Products Processing of Zhejiang Province, Collaborative Innovation Center of Seafood Deep Processing, Institute of Seafood, Zhejiang Gongshang University, Hangzhou 310012, China; 19020080042@pop.zjgsu.edu.cn (D.C.); 21020080074@pop.zjgsu.edu.cn (Y.J.); 21020080088@pop.zjgsu.edu.cn (D.H.); yellowleaf@zjgsu.edu.cn (J.Y.); dzy@mail.zjgsu.edu.cn (Z.D.)

**Keywords:** brown algae, *Sargassum fusiforme*, *Laminaria japonica*, *Undaria pinnatifida*, fucoxanthin, separation, elution-extrusion, countercurrent chromatography

## Abstract

A method for batch preparation of fucoxanthin from brown algae was established, which possessed the advantages of high yield and high purity. The ultrasonic-assisted extraction method was used to obtain a crude extract from *Sargassum fusiforme* as the separation sample. Then the crude extract was separated by elution-extrusion countercurrent chromatography. The optimum preparation conditions of fucoxanthin were determined as follows: *n*-hexane-ethanol-water (20:9:11, *v*:*v*:*v*) as a two-phase solvent system, the mobile phase flow rate was 5 mL min^−1^, the revolution speed was 800 r min^−1^, the loading capacity was 60 mg 10 mL^−1^ and the temperature was 25 °C. By this method, 12.8 mg fucoxanthin with a purity of 94.72% was obtained from the crude extract of *Sargassum fusiforme*. In addition, when the loading capacity was 50 mg 10 mL^−1^, the purity of fucoxanthin reached 96.01%. Two types of by-products, chlorophyll and pheophytin, could also be obtained during the process of separation. This optimal method was further applied to separate fucoxanthin from *Laminaria japonica* and *Undaria pinnatifida*, and 6.0 mg and 9.7 mg fucoxanthin with a purity of 96.24% and 92.62% were acquired, respectively. Therefore, it was demonstrated that the preparation method of fucoxanthin established in this study had an applicability to brown algae, which improved the utilization value of raw materials.

## 1. Introduction

In recent decades, research on marine resources has been increasing with people considering land-use and environmental protection. As the most important producer, seaweed has abundant resources. There are various types of algae divided into three categories: red algae, brown algae and green algae. Brown algae is the second largest group of marine algae, named as its color is brown. It includes approximately 250 genera and 1500 species [1,2]. The common edible brown algae mainly include *Laminaria japonica*, *Undaria pinnatifida*, and *Sargassum fusiforme* (Figure 1).

Fucoxanthin is a natural carotenoid, belonging to the lutein class. It usually exists in algae in the form of fucoxanthin-chlorophyll protein complex, which plays a role in light capture and light transmission [3]. Fucoxanthin is most abundant in algae of Phaeophyceae and Diatoms. According to statistics, the content of carotenoids in nature is abundant. For instance, fucoxanthin in brown algae accounts for more than 10% of the total carotenoid production in nature [4], lutein accounts for 5% of marigold petal [5], while astaxanthin accounts for 4% in *Haematococcus pluvialis* [6]. Most modern medical drugs are synthetic drugs, which have potential safety hazards, such as toxicity and side effects [7]. Moreover, the defects in synthesis process, production cost and environmental protection restrict the further development of synthetic drugs. This highlights the advantages of non-toxic and harmless natural compounds [8,9,10]. Fucoxanthin has a unique structure (Figure 1), including an ethylenic bond and multiple oxygen-containing functional groups such as epoxy, hydroxyl, carbonyl and carboxyl groups [11]. This unique structure gives fucoxanthin a variety of beneficial biological activities related to human health, such as antioxidant, anti-obesity, anti-diabetic, anti-cancer, anti-inflammatory, etc. [12,13,14,15,16]. Therefore, fucoxanthin has great potential to replace synthetic drugs in the treatment of certain diseases.

However, one of the main problems inhibiting wider commercial use of fucoxanthin is the low efficiency ratio and cost associated with the extraction and purification processes used nowadays. Fucoxanthin has similar physical and chemical properties as other carotenoids. Therefore, in the process of extracting fucoxanthin, it is usually accompanied by the dissolution of other carotenoids and chlorophyll. This undoubtedly increases the difficulty of purifying fucoxanthin. At present, thin layer chromatography and silica gel column chromatography are widely used in the purification of carotenoids [17,18,19]. Although their separation effect is good, they have some shortcomings, such as inconvenient operation, large solvent consumption, long separation time, and less sample loading [20,21]. High-speed countercurrent chromatography (HSCCC) is an efficient preparation technology that uses the principle of liquid–liquid distribution to separate the mixture. That is, without using a solid-phase carrier, the target compound is continuously distributed between the immiscible liquid–liquid phases to achieve the purpose of separation [22]. HSCCC has the advantages of flexible elution method, low solvent consumption, large loading capacity, high product purity, simple operation, etc. [23]. According to reports, HSCCC has been widely used to separate carotenoids from pigment mixtures, and has achieved good results, such as astaxanthin [23], lycopene [24], anthocyanins [25], lutein [26], etc. At present, the HSCCC method for separating fucoxanthin from brown algae is limited to the earlier study for obtaining fucoxanthin with low yield and low purity [27]. Therefore, it is meaningful to separate fucoxanthin effectively. Elution-extrusion countercurrent chromatography (EECCC) is a kind of countercurrent chromatography that has the advantage of recovering all samples through one elution. This method can quickly elute the solute retained on the column, thereby reducing solvent consumption and separation time [28].

In this paper, we tried to use EECCC technology to separate and purify fucoxanthin from the crude extract of *S. fusiforme*. HPLC and UV-Vis spectrophotometer methods had been applied for determination of the fractions separated by EECCC. The applicability of the established EECCC method for the separation of fucoxanthin from other browns algae was validated.

## 2. Results and Discussions

### 2.1. Selection of the Two-Phase Solvent System

The determination of a suitable two-phase solvent system is the first step of EECCC separation, which can provide an ideal K value for the target compound. According to reports, the K value should be in the range of 0.5–2.0 to obtain a suitable running time and effective separation. If the K value is too low, solutes will accumulate and be eluted in the early stage, which may cause loss of peak resolution. Higher K values, although the separation effect is better, produce a wider peak and prolong the separation time [23]. In this study, *n*-hexane-ethanol-water was used as the solvent system, and the K value in the solvent system with different volume ratios was determined by HPLC. From Table 1, the K value gradually increases with the increase in the proportion of water. This shows that the increase in the proportion of water in the solvent system was beneficial to the separation of fucoxanthin. The K values of 20:10:10 and 20:9:11 were 0.65 and 1.74, respectively, which were all in the appropriate range. In order to ensure effective separation, 20:9:11 was selected as the solvent system for separating fucoxanthin.

### 2.2. Optimization of Elution-Extrusion Countercurrent Chromatography

Retention of stationary phase (RSP) is an important factor affecting the separation of EECCC. The volume of stationary phase retained in the column has a great influence on peak resolution. The higher the retention level, the better the peak resolution [23]. In general, when the RSP is higher than 40%, a better separation effect can be obtained. When it is lower than 40%, a large amount of stationary phase will be lost, which can significantly reduce the separation effect [29]. According to previous studies, RSP is affected by the changes of flow rate of the mobile phase and revolution speed [22]. Therefore, the two conditions were optimized in this study.

#### 2.2.1. Effect of the Flow Rate of the Mobile Phase on the Retention of the Stationary Phase

The effects of different flow rates of the mobile phase on RSP were compared at a fixed rotational speed of 750 r min^−1^. According to Figure 2A, RSP gradually decreased with the increase of flow rate, and there was an obvious linear relationship between them. The regression equation was obtained by linear fitting (Equation (1)). Englert et al. also found a good linear relationship between the mobile phase flow rate and RSP [26]. When the flow rate was lower than 4 mL min^−1^, the values of RSP were higher (>70%), but this also meant that a longer separation time was required. When the flow rate was 5 mL min^−1^, the RSP could also reach 69.05%. In order to shorten the separation time, 5 mL min^−1^ was selected as the flow rate of the mobile phase in this study.
(1)$$y=-1.67\;x+77.18{\;(R}^{2}=0.9967)$$

#### 2.2.2. Effect of Revolution Speed on the Retention of the Stationary Phase

The effects of different revolution speeds on RSP were compared at a fixed mobile phase flow rate of 4 mL min^−1^. The results are shown in Figure 2B. With the increase of revolution speed, the value of RSP increased gradually, which was consistent with the trend observed by previous research [30]. The value of RSP was above 70% when the speed was higher than 750 r min^−1^. When the speed exceeded 800 r min^−1^, the rising trend of RSP slowed down. The system pressure will increase gradually with the increase of revolution speed [31]. The higher speeds may produce excessive sample band broadening by intense pulse of the column because of elevated pressure. On the other hand, when the pressure exceeds the critical point, the pipeline may rupture. Therefore, 800 r min^−1^ was selected as the best revolution speed.

### 2.3. Elution-Extrusion Countercurrent Chromatography Separation of Fucoxanthin

After the above optimization, the basic conditions for purifying fucoxanthin from the crude extract of *S. fusiforme* were determined. The concentration of the injected sample, which is the loading capacity, will also have a great influence on the separation effect. Excessive capacity will result in loss of stationary phase and low resolution, while insufficient capacity will reduce the yield of target compounds [32]. Therefore, the effect of different loading capacity on the separation effect was studied. The result is shown in Figure 3. Through HPLC analysis, fucoxanthin was detected in the fractions during 78–116 min with a higher purity. This also indicated that fucoxanthin could be eluted during this period. When the loading capacity was 50 mg, the peaks were separated from each other with higher resolution, followed by 60 mg. For 80 and 100 mg, the peaks representing fucoxanthin appeared in the form of shoulder peaks with lower resolution, and some peaks could not be distinguished.

After the separation, the peak fractions corresponding to fucoxanthin were combined and dried to obtain purified fucoxanthin. The content and purity of fucoxanthin were detected through HPLC. The results are shown in Table 2. For *S. fusiforme*, with the increase of loading capacity, the yield of fucoxanthin gradually increased, but the purity gradually decreased. When the loading capacity was 50 mg, 11.3 mg of fucoxanthin could be recovered, and its purity was as high as 96.01%. When the sample load was increased to 100 mg, although the yield of fucoxanthin increased significantly, the purity was only 85.37%. Therefore, 60 mg was selected as the best loading capacity, because it could obtain the high demand of purity as well as appropriate yield at the same time.

Eventually, the optimal EECCC conditions were obtained: *n*-hexane-ethanol-water (20:9:11, *v*:*v*:*v*) as a two-phase solvent system, the mobile phase flow rate was 5 mL min^−1^, the revolution speed was 800 r min^−1^, the loading capacity was 60 mg 10 mL^−1^ and the temperature was 25 °C. The precision and repeatability of this method were validated and the results (Table 3) showed that they were all less than 6%, which proved the feasibility of the established method.

In order to verify the general applicability of the optimal EECCC separation method established in this study, the method was then applied to separate fucoxanthin from other brown algae, such as *L. japonica* and *U. pinnatifida*. From Figure 3, the chromatograms of *L. japonica* and *S. fusiforme* were similar. The separation of 60 mg crude extract of *L. japonica* achieved good results, and the resolution of the fucoxanthin peak was very high. The yield of fucoxanthin was 6.0 mg and the purity was 96.24%. For the separation of 60 mg crude extract of *U. pinnatifida*, other peaks appeared in the chromatogram, and the resolution of the fucoxanthin peak was lower. However, the analysis of the fractions during 78–116 min showed that the yield of fucoxanthin was 9.7 mg, and the purity remained above 92%. Therefore, the preparative separation method of fucoxanthin established in this study was an efficient method with universal applicability to brown algae.

### 2.4. Analysis of HPLC and UV-Vis Full Scan

The best separation effect was obtained when the injection concentration of the crude *S. fusiforme* extract was 50 mg 10 mL^−1^, and the peak resolution was the highest. Therefore the fractions obtained under this condition were analyzed to determine their composition. As shown in Figure 4, the mixture was divided into four components, which correspond to fractions F_1_, F_2_, F_3_, and F_4_, respectively. The first three fractions were obtained through normal elution, while F_4_ was pushed out after changing the mobile phase around 115 min. According to the diagram in Figure 4, the colors of the four fractions were green, colorless, yellow, and brown, which meant they contain different pigments.

The crude extract, fucoxanthin standard and four fractions (F_1_–F_4_) were analyzed by HPLC at 449 nm. Five compounds can be found from the crude extract, and their retention times were 2.18, 3.77, 5.75, 7.72 and 14.09 min, respectively (Figure 5A). According to Figure 5A, it could be found that F_1_ contained three compounds C_1_, C_2_ and C_3_. F_3_ contained compound C_4_, which was fucoxanthin, and a small amount of compound C_5_. At the wavelength of 449 nm, no compound was clearly detected in F_2_. Combined with the color of the eluate, it was speculated that the pigment content in F_2_ was not high, and there may be non-pigment compounds. Li et al., 2016 used HSCCC to separate cocoa bean polyphenols into different fractions, which contained procyanidins ranging from monomers to pentamers, and the major individual procyanidins were further separated from these fractions by preparative HPLC [33]. The presence of compound C_5_ was the main reason for reducing the purity of fucoxanthin. In a subsequent study, preparative HPLC can also be used to separate fucoxanthin and compound C_5_ to further improve the purity of fucoxanthin.

In order to further determine the compounds in the fractions, this study continued to use a UV-Vis spectrophotometer to do full-scan spectral analysis of the fractions. From Figure 5B, the characteristic absorption peak of the fucoxanthin standard spanned the range of 400–500 nm, and the maximum absorption wavelength was about 449 nm. The absorption peak of F_3_ was similar to the fucoxanthin standard, which proved that F_3_ contained high-purity fucoxanthin. With reference to previous studies [21,34,35], the pigment corresponding to each absorption peak in the spectrum of the crude extract can be determined. The absorption peaks at 415 nm and 432 nm were produced by pheophytin and chlorophyll a, respectively. The absorption peak of chlorophyll c was between 550–650 nm. Both chlorophyll and pheophytin absorb near 665 nm. The maximum absorption peak of F_1_ was at 443 nm, and F_4_ had two obvious absorption peaks at 408 and 665 nm respectively. This showed that through EECCC separation, chlorophyll was eluted in the early stage and was detected in F_1_. Pheophytin was pushed out in the later stage and was detected in F_4_. When the two types of pigments were separated for UV detection, there was no interference between the two, so the absorption peaks were shifted to both sides respectively. Combined with Figure 5, the pigment in F_1_ was mainly green chlorophyll, and the pigment in F_4_ was mainly brown pheophytin. According to the current information, the chemical composition of F_2_ failed to be determined, but it would be identified by our next experiment.

## 3. Materials and Methods

### 3.1. Materials and Reagents

Fresh *S. fusiforme* was purchased from Dongtou (Zhejiang, China), *L. japonica* and *U. pinnatifida* were purchased from local supermarkets (Zhejiang, China). All these samples were dried in a vacuum oven at 55 °C to a constant weight, then grounded into powder, and stored under dark conditions for later use. Analytical grade ethanol, acetone, ammonium acetate, and *n*-hexane were purchased from Zhejiang Changqing Chemical Ltd. (Hangzhou, China). Methanol and acetonitrile utilized for HPLC were chromatographic reagent obtained from Merck (Darmstadt, Germany). Fucoxanthin standard (≥95.0%) was purchased from Sigma-Aldrich (Shanghai, China). The ultrapure water was purified by Milli-Q system (Bedford, MA, USA), with a resistivity of 18.2 MΩ/cm.

### 3.2. Preparation of Crude Extract

The preparation method of the crude extract referred to the extraction method of fucoxanthin in our laboratory. A certain amount of raw material was weighed and mixed evenly with the extraction solvent ethanol/acetone (*V*/*V*, 3:1), and the liquid–solid ratio was 20 mL g^−1^. The mixture was kept in the dark for 2 h. After preheating, the VCX500 ultrasonic breaker (Sonics & Materials INC, Newtown, CT, USA) was used as an assistant to extract fucoxanthin [36,37]. The extraction conditions were as follows: temperature of 65 °C, time of 20 min, amplitude of 40%, running for 10 s, with an interval of 10 s. The obtained extract was centrifuged at 4 °C and 8000 r min^−1^ for 10 min (Sorvall RC 6 Plus high-speed refrigerated centrifuge, Thermo, Waltham, MA, USA), and the supernatant was collected. The residue was extracted twice, the supernatant was combined, and the solvent was evaporated by rotary distillation (RE-3000 Rotary Evaporator, Shanghai Yarong Biochemical Instrument Factory, Shanghai, China). The dried crude extract was stored in the refrigerator (4 °C) for the next experiment.

### 3.3. Quantitation of Fucoxanthin by High-Performance Liquid Chromatography

HPLC detection conditions were slightly modified based on existing methods for detecting fucoxanthin [38]: e2695 HPLC (Waters, Milford, MA, USA), Welchrom^®^ C18 column (250 mm × 4.6 mm, 5 μm i.d.) (Welch Materials Inc., Shanghai, China), detection wavelength was 449 nm; the mobile phase included acetonitrile: methanol: 0.1% ammonium acetate aqueous solution (*v*:*v*:*v*, 75:15:10) with isocratic elution mode, flow rate of 1 mL min^−1^, injection volume of 20 μL, column temperature of 30 °C, and the detection time was 20 min.

The fucoxanthin standard was prepared into solutions with different concentrations including 1, 2, 4, 8, 16, and 32 μg/mL, which were analyzed by HPLC. Linear regression analysis was made between the peak area of fucoxanthin and its concentration. The quantitative equation for fucoxanthin was obtained (Equation (2)).
(2)y=176603 x− 105594
where y is the concentration of fucoxanthin (μg/mL) in the sample solution and x is the peak area of fucoxanthin detected by HPLC.

### 3.4. Selection of the Two-Phase Solvent System

*n*-hexane-ethanol-water was selected as the two-phase solvent system for the separation of fucoxanthin, which referred to the method of purification of lutein [39]. The value of the distribution coefficient (K) is an important index to evaluate the applicability of a solvent system [23]. The K value of fucoxanthin in the solvent system was determined according to the following method:

*n*-hexane, ethanol and water were mixed according to the set volume ratio (20:11:9, 20:10:10, 20:9:11, 20:8:12, and 20:7:13). After equilibrium and stratification, 2 mL upper phase and 2 mL lower phase were put into a 10 mL centrifuge tube. Next, 0.5 mg crude extract was completely dissolved by the above mixed solvent. After fully dissolving, 1 mL upper phase and 1 mL lower phase were filtered with 0.45 μm nylon membrane, respectively. The peak area of fucoxanthin in the two phases was determined by HPLC (specific in Section 3.3). According to Equation (3), the K value of fucoxanthin in the solvent system was calculated.
(3)$$K=\frac{A_{1}}{A_{2}}$$
where A_1_ and A_2_ are the peak areas of fucoxanthin in the upper and lower phases, respectively.

### 3.5. Preparation of Elution Solvent and Sample Solution

After acquiring the appropriate two-phase solvent system, the solvents was added into the separating funnel according to the volume ratio and mixed completely. The upper and lower phases were separated after stratification. The two phases were ultrasonically degassed in a water bath for 30 min (VCX750 Ultrasonic Cleaner, Sonics & Materials INC., Newtown, CT, USA) and utilized as the elution solvent. Crude extracts of different qualities (50, 60, 80, and 100 mg) were dissolved in the solvent system consisting of 5 mL upper phase and 5 mL lower phase to prepare sample solutions with different concentrations.

### 3.6. Optimization of Elution-Extrusion Countercurrent Chromatography

In this study, EECCC was used to separate fucoxanthin. The elution mode was reversed phase from head to tail, that is, the upper phase was utilized as the stationary phase while the lower phase was the mobile phase. Before injection, the low temperature thermostat was opened and the temperature was set to 25 °C. The stationary phase was pumped into the column at a flow rate of 25 mL min^−1^. After the whole pipeline was filled with the stationary phase, the main engine (TBE-300C HSCCC, Shanghai Tongtian Biotechnology Co., Ltd., Shanghai, China) was turned on and the revolution speed was adjusted to the set value (650, 700, 750, 800, and 850 r min^−1^) in a positive direction. After the revolution speed was stable, the flow rate was adjusted to the set value (2, 3, 4, 5, and 6 mL min^−1^). The stationary phase was replaced by the mobile phase, and the stationary phase flowing out from the outlet of the detector was collected by volumetric cylinder and recorded as V_S_. Through the UV detector, when the baseline reached equilibrium, it indicated that the entire system reached hydrodynamic equilibrium. At this time, the retention of the stationary phase (RSP) could be calculated by Equation (4), which was an index for selecting the optimal flow rate and revolution speed.
(4)$$\mathrm{RSP}=\left(1\;-\;\frac{V_{s}}{V_{t}}\right)\;\times \;100$$
where V_s_ (mL) is the volume of the stationary phase in the volumetric cylinder, and V_t_ is the total volume of the chromatographic column (310 mL).

After the EECCC system reached equilibrium, the sample solution was injected into the chromatographic column from the injection valve, and the record button of the chromatography workstation was immediately turned on to record the UV absorption chromatogram. The detection wavelength was set to 405 nm. During the separation process, the liquid flowing out from the outlet of the detector was collected by an automatic collector after injection. Preliminary experiments showed that the fucoxanthin was completely eluted in about 115 min. Therefore, the mobile phase was replaced by the stationary phase again at 115 min, and the elution continued. When all the pigments were pushed out, that is, the eluate no longer had color, the elution was finished.

### 3.7. Verification of the Optimal EECCC Method

The EECCC method was verified from three aspects: repeatability, precision and applicability. The repeatability and precision were performed according to previous methods [40], while the applicability was verified by applying the optimal EECCC method to separate fucoxanthin from other brown algae: *Laminaria japonica* and *Undaria pinnatifida*.

### 3.8. Detection of Elution-Extrusion Countercurrent Chromatography Fractions

According to the countercurrent chromatogram, the peak fractions corresponding to fucoxanthin were combined to obtain purified fucoxanthin by rotary distillation; 0.2 mg purified fucoxanthin was redissolved with 10 mL ethanol. The content of fucoxanthin was detected by HPLC to obtain the peak area. The peak area was measured by Equation (2) to acquire the concentration of fucoxanthin (C_F_), while the purity of fucoxanthin (W_F_) was calculated by Equation (5).
(5)$$W_{F}=\frac{{10C}_{F}}{0.2\;\times \;1000}\;\times \;100\%$$

In order to determine the other fractions separated by EECCC, HPLC was used to analyze the fractions. They were also scanned by UV-vis spectrophotometer with the wavelength range of 350–750 nm (Evolution 60 S, Thermo Scientific, Shanghai, China), and ethanol was used for zero adjustment.

### 3.9. Statistical Analysis

All experiments were repeated three times. The results were reported as means ± standard deviation (SD). Duncan’s multiple range test (5% of confidence level) was carried out using SPSS v21.0. According to Duncan’s test, whether there were significant differences between the data was judged.

## 4. Conclusions

In this study, an efficient preparation method was established to successfully separate high-purity fucoxanthin from brown algae using EECCC. By this method, 12.8, 6.0, and 9.7 mg of fucoxanthin were separated from the pigment mixture of 60 mg *S. fusiforme*, *L. japonica*, and *U. pinnatifida*, respectively, and the purity was 94.72%, 96.24%, and 92.62%, respectively. In addition, this method can separate two other major types of pigments (chlorophyll and pheophytin) from crude extract at different time periods. Overall, the developed method might serve as a reference for contributing to the development of a new mode of fucoxanthin production based on brown algae. In follow-up studies, efforts will be made to increase the recovery of the two by-products of chlorophyll and pheophytin to improve the value of this method as well as identify what kind of substance F_2_ is. We will also look for suitable modifiers to increase the separation speed.

## Figures and Tables

**Figure 1 marinedrugs-20-00257-f001:**
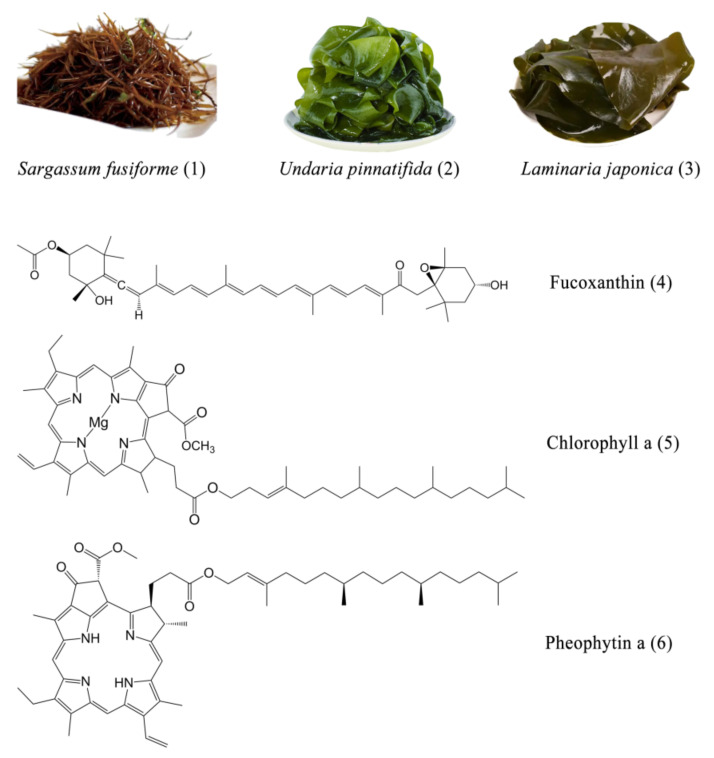
Physical map of three brown algae: *Sargassum fusiforme* (**1**), *Undaria pinnatifida* (**2**), *Laminaria japonica* (**3**), as well as the chemical structures of fucoxanthin (**4**), chlorophyll a (**5**) and pheophytin a (**6**).

**Figure 2 marinedrugs-20-00257-f002:**
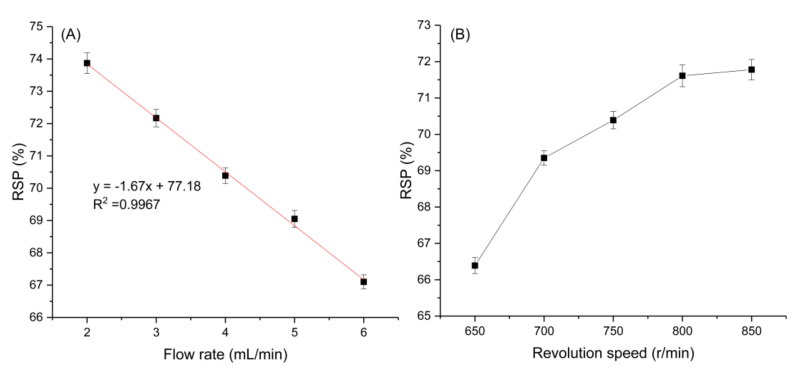
(**A**) Effects of flow rate of mobile phase on the retention of stationary phase (rotational speed: 750 r min^−1^); (**B**) Effects of rotational speed on the retention of stationary phase (flow rate: 4.0 mL min^−1^).

**Figure 3 marinedrugs-20-00257-f003:**
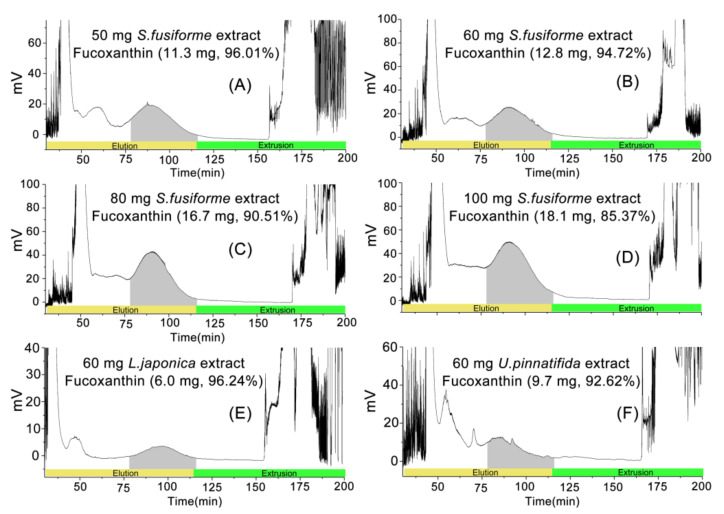
(**A**–**D**): Effect of loading capacity (50 mg, 60 mg, 80 mg, 100 mg) on the separation of fucoxanthin from *Sargassum fusiforme* crude extracts; (**E**) and (**F**): The application of the optimal method to separate fucoxanthin from *Laminaria japonica* and *Undaria pinnatifida*, respectively. The gray area was the time that fucoxanthin was being collected.

**Figure 4 marinedrugs-20-00257-f004:**
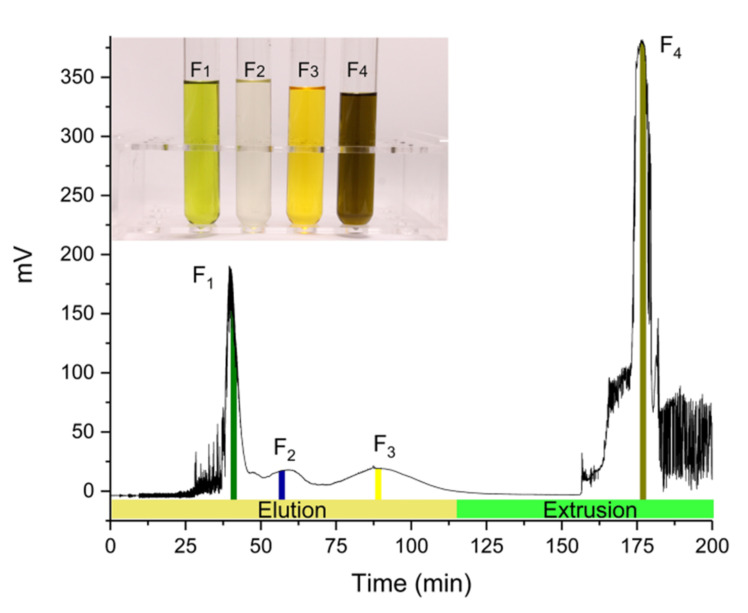
Elution-extrusion countercurrent chromatography separation chromatogram of 50 mg crude *Sargassum fusiforme* extract. F_1_, F_2_, F_3_, and F_4_ correspond to fractions of 40–42, 56–58, 88–90 and 176–178 min, respectively.

**Figure 5 marinedrugs-20-00257-f005:**
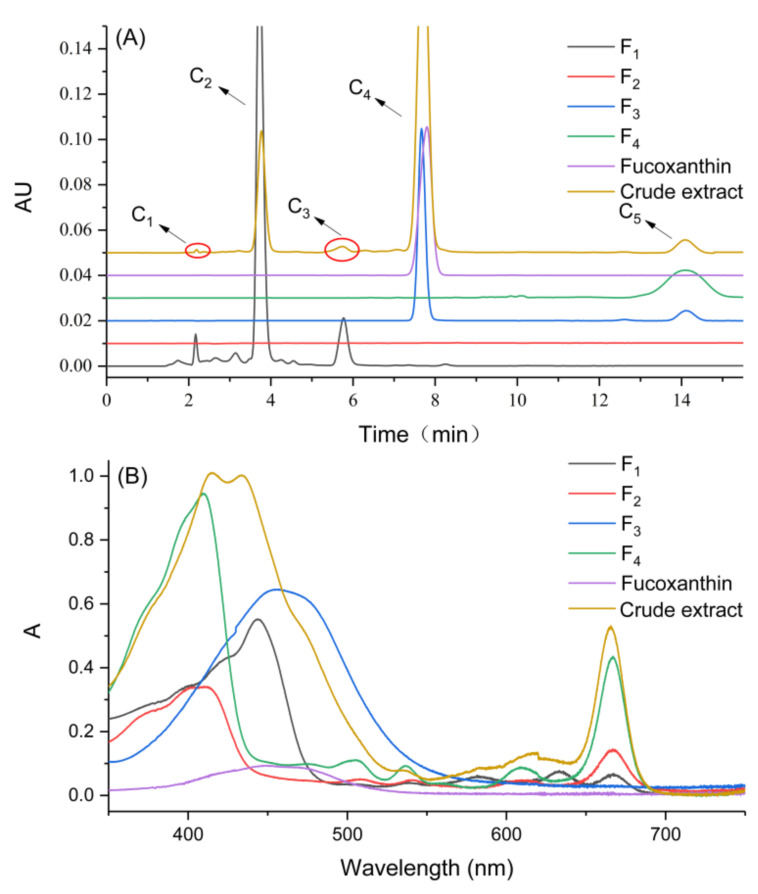
Analysis of fractions (F_1_–F_4_) separated by EECCC, *Sargassum fusiforme* crude extract, and fucoxanthin standard through two kinds of methods: (**A**) HPLC chromatogram; (**B**) UV-Vis spectra.

**Table 1 marinedrugs-20-00257-t001:** K values of *S. fusiforme* fucoxanthin in different solvent systems.

Solvent System	Volume Ratios	K
*n*-hexane-ethanol-water	20:11:9	0.23 ± 0.09
	20:10:10	0.65 ± 0.08
	20:9:11	1.74 ± 0.11
	20:8:12	2.28 ± 0.12
	20:7:13	3.61 ± 0.14

**Table 2 marinedrugs-20-00257-t002:** The effect of different loading capacity on the yield and purity of fucoxanthin.

Samples	Loading Capacity (mg/10 mL)	Yield (mg)	Purity (%)
*S. fusiforme*	50	11.3 ^b,c,^*	96.01 ± 1.80 ^A^ ^**^
	60	12.8 ^b^	94.72 ± 1.47 ^A^
	80	16.7 ^a^	90.51 ± 1.53 ^C^
	100	18.1 ^a^	85.37 ± 1.63 ^D^
*L. japonica*	60	6.0 ^d^	96.24 ± 1.58 ^A^
*U. pinnatifida*	60	9.7 ^c^	92.62 ± 1.77 ^B^

* The different lowercases indicate that there was a statistically significant difference (*p* < 0.05) between the different yields.** The different capital letters indicate that there was a statistically significant difference (*p* < 0.05) between the different purity.

**Table 3 marinedrugs-20-00257-t003:** The precision and repeatability of the optimal EECCC method.

	Repeatability (RSD * %, *N* = 3)	Precision (RSD%, *N* = 6)
Yield (mg)	5.2	5.9
Purity (%)	1.6	2.3

* RSD was relative standard deviation.

## Data Availability

Not applicable.

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
