# Peer review of "One-Step Preparative Separation of Fucoxanthin from Three Edible Brown Algae by Elution-Extrusion Countercurrent Chromatography"

_marinedrugs, 2022, doi:10.3390/md20040257_

Round 1
Reviewer 1 Report
In my opinion, the work was well performed and the subject is of important interest. The separation process was well succeeded in high purity level and showed good resolution between the target compounds.
I suggest minor revisions to the authors:
1- the abstract should be rewritten in order to describe better the several experiments performed by the authors. As it is, it is it do not reflect the all that was done. Many kinds of experiments, and target compounds isolation in purity level above the described in the abstract (some fucoxanthin was isolated in 96,24% purity lvl). So pls, redo it giving more worth to the obtained data.
2- part of the introduction is related to the results and should be replaced (
"In that study, n-hexane-ethyl acetate-ethanol-water was used as the solvent system. After one separation, the highest yield of fucoxanthin was 2.88 mg with a purity of 89.3%, and the highest purity was 94.8% with a 73 yield of 0.89 mg. After the fucoxanthin was separated, the separation process was immediately stopped, and the other fractions were not analyzed." and "By comparing the partition coefficients, the appropriate solvent system was selected first. Then the flow rate of the mobile phase, revolution speed and loading capacity of EECCC were optimized. The components of different fractions were also analyzed by HPLC and UV-vis spectrophotometer. Finally, the applicability of the established EECCC method for the separation of fucoxanthin from other browns algae was verified."
3- The references must to be checked and rewritten according to the journal rules. Some are rewritten as:
"Appl Biochem Micro 2010, 46, 267-273" and some
"Res. J. Pharm. Technol 2015, 8, 1379-1382."

Reviewer 2 Report
Fucoxanthin is an active ingredient of marine algae belonging to the group of carotenoids (which in itself proves some antioxidant properties). It occurs in macro- and microalgae such as: Undaria pinnatifida, Laminaria japonica, Phaedactylum tricornutum and Cylindrotheca closterium. Due to the unstable structure of fucoxanthin, it is attributed the aforementioned antioxidant properties. Its action has an extensive effect on the body in the form of anti-diabetic, hepatoprotective, anti-inflammatory, anti-cancer, and cardiovascular protection. However, its main effect is the prevention of obesity and the resulting diseases of civilization. This study developed an efficient preparation method to efficiently separate high purity fucoxanthin from brown algae using EECCC.
The authors made a literature review based on the latest publications (mainly from the last 10 years). The experiment was properly planned and carried out. The assumptions of the work have been achieved. The obtained research results were correctly discussed.
According to the reviewer, the authors should provide information on the precision and repeatability of the analytical method used.
Reviewer 3 Report
An efficient preparative method for fucoxanthin from brown algae was elaborated. The ultrasonic-assisted extraction from Sargassum fusiforme provided a crude sample that was then subjected to elution-extrusion countercurrent chromatography. The optimum conditions of the procedures were carefully evaluated. The purity of fucoxanthin was 95% when obtained from Sargassum fusiforme. Authors proved that the preparation method for fucoxanthin is of more general value in case of brown algae. Two by-products, chlorophyl and pheophytin, could also be made available.
The results justify publication in Marine Drugs after the following minor revisions:
- in Figure 1, the important structures are too small, pls enlarge.
- Au-s write: „According to statistics, the content of fucoxanthin in brown algae accounts for more than 10% of the total carotenoid production in nature [4].” Would it be possible to say a little bit more about the conversion of the extract to carotenoid?
- It is indeed a green chemical approach to extract chemicals rather than synthesizing them. It would be advicable to cite a few books on extractive approaches. The Royal Society of Chemistry definitely has such book(s) in its good series.
- in Figure 3, the pictures A-F are too small, pls enlarge.
- the weight quantities should not be given to two decimals, e.g. 11.25 mg – pls correct this throughout.
- this referee considers it excessive to give a yield as 25 ± 0.59 mg, the „±” part is unnecessary in Table 2. Pls correct throughout. Regarding the purity, it is OK to give the range.
